# The Role of Grass in the Epidemiology of a Phytoplasma Disease Affecting Trees and Other Plants of the Sabana de Bogotá, Colombia

**DOI:** 10.3390/microorganisms13050967

**Published:** 2025-04-23

**Authors:** Liliana Franco-Lara, Aura Cristina Campo-Garnica, Iris Calanit French, Cindy Julieth Solano, Maria Nathalia Vargas

**Affiliations:** 1Faculty of Basic and Applied Sciences, Universidad Militar Nueva Granada, Cajicá 110131, Colombia; 2Independent Researcher, Bogotá 110121, Colombia; acampogarnica@gmail.com; 3Alkemist Labs, Garden Grove, CA 92841, USA; iris@alkemist.com; 4Independent Researcher, Santa Marta 470008, Colombia; cindysolano16@gmail.com; 5National Health Institute, Bogotá 111321, Colombia; mvargasf@ins.gov

**Keywords:** group 16SrI, group 16SrVII, epidemiology

## Abstract

‘*Candidatus* Phytoplasma asteris’ and ‘*Candidatus* Phytoplasma fraxini’ infect at least nine species of trees, potato, and strawberry crops in the Sabana de Bogotá. We analyzed the epidemiological implications of the presence of these phytoplasma species in trees, grass, and weeds in the Sabana de Bogotá, as well as in Cicadellidae insects. Both phytoplasmas were detected in symptomatic trees of *Salix humboldtiana* and *Sambucus nigra*, and in asymptomatic grass *Cenchrus clandestinus* and 13 weed species. ‘*Ca*. P. asteris’ and ‘*Ca*. P. fraxini’ sequences of the 16S rRNA gene obtained from positive samples were compared with sequences from plants and insects of the Sabana de Bogotá. In each case, the sequence identity of this gene suggested that closely related strains of each species circulate in the environment, infecting plants of many families and several Cicadellidae species. *Ce. clandestinus* plays a key role in the epidemiology of the disease since it is a host of both phytoplasmas, of two known insect vectors, and of other Cicadellidae. *Ce. clandestinus* is extensively distributed in urban and rural areas. Since management efforts are hampered by the practical impossibility to remove *Ce. clandestinus* from the ecosystem, different strategies are needed to manage this disease.

## 1. Introduction

Phytoplasmas are bacteria of the order Acholeplasmatales, class Mollicutes [1]. They are associated with hundreds of plant diseases in many parts of the world. Phytoplasmas lack a cell wall and are obligate parasites that live in the phloem of their plant hosts or in the hemolymph of hemipteran insects [2,3]. They are considered unculturable microorganisms, because they are fastidious, and attempts to produce axenic cultures of phytoplasma have failed [1]. Phytoplasmas are transmitted mainly by insect vectors of the order Hemiptera, but also by vegetative material, grafting, and occasionally by seeds [3]. Phytoplasmas are associated with symptoms such as yellowing, disturbances in the plant growth patterns, and generation of phyllody and virescence in flowers, among others [2]. Their taxonomic classification relies on the 16S rRNA gene sequence and currently they are classified as part of the provisional genus ‘*Candidatus* Phytoplasma’ [4,5].

The Sabana de Bogota is a high plateau of approximately 1.395 Ha located at 2600 m of altitude in the Eastern Cordillera de Los Andes, in central Colombia. It is a part of the Cundinamarca department, and it includes urban areas such as Bogotá, the largest city in the country, and 20 smaller towns. These urban areas are intermixed with rural areas in which crops such as potatoes, strawberries and other fruits, corn, and fresh cut flowers are cultivated. Additionally, native Andean Forest strongholds persist in some areas, a reminder of the original ecosystem of the Sabana. In addition to native plant species, a large number of introduced species are currently found in this plateau including trees, bushes, grass, and cultivated and ornamental plants.

In the Sabana de Bogotá, ‘*Candidatus* Phytoplasma asteris’ (group 16SrI) and ‘*Candidatus* Phytoplasma fraxini’ (group 16SrVII) have been associated with a disease that affects several tree species in urban and rural areas of the Sabana de Bogotá. Diseased trees include *Acacia melanoxylon* R.Br. (Fabaceae), *Croton* spp. (Euphorbiaceae), *Fraxinus uhdei* (Wenz.) Lingelsh. (Oleaceae), *Eugenia neomyrtifolia* Sobral (Myrtaceae), *Liquidambar styraciflua* L. (Altingiaceae), *Magnolia grandiflora* L. (Magnoliaceae), *Pittosporum undulatum* Vent. (Pittosporaceae), *Populus nigra* L. (Salicaceae), and *Quercus humboldtii* Bonpl. (Fagaceae) [6,7,8,9,10,11]. Additionally, these two species of phytoplasmas infect *Solanum tuberosum* and *Solanum phureja* (Solanaceae) potato plants [12] and *Fragaria* X *ananassa* (Rosaceae) strawberry crops [7].

In this work, we report the occurrence and analyze the epidemiological implications of the presence of ‘*Ca*. P. asteris’ and ‘*Ca*. P. fraxini’ in trees, grass, and weeds in the Sabana de Bogotá. *Salix humboldtiana* Willd. (Salicaceae) is a native tree of the Americas and it grows naturally from Mexico to Argentina. It has ornamental and ecological value with adult trees growing as high as 25 m. It has arched crowns with light green foliage, and thin alternated leaves of approximately 10 cm [13]. It is frequently found growing near rivers, streams, lakes, and zones of temporary floodings [14]. *Sambucus nigra* L. (Adoxaceae) is an introduced species originally from Europe, northeast Africa, and southeast Asia. In Colombia, it is used as an ornamental plant, as a living fence, and is frequently found growing on median strips, sidewalks, and parks of cities. It is a shrub, usually 4 to 6 m high with round, dense crowns and wrinkled bark [13]. Specimens of these two tree species growing in the Sabana de Bogotá show symptoms that suggest phytoplasma infection. *Cenchrus clandestinus* (Hochst. ex Chiov.) Morrone (kikuyu grass) was introduced from Africa to Colombia as fodder for cattle about 100 years ago. Since then, it was naturalized in Colombia and at present it is widely distributed in rural areas of the Sabana de Bogotá and in many cattle production areas of Colombia. Furthermore, it became the predominant ornamental grass in many cities located over 1500 m of altitude and it grows in in parks, gardens, and median strips in cities and towns of the Sabana de Bogotá. It behaves as a weed and has displaced many of the original grass species of the area [15]. At least 30 species of leafhoppers thrive on *Ce. clandestinus* in the Sabana de Bogotá [16,17], including 2 known phytoplasmas vectors for the area, *Amplicephalus funzaensis* Linnavuori 1968 and *Exitianus atratus* Linnavuori 1959 (both Hemiptera: Cicadellidae) [18]. Additionally, samples of 35 weed species of different botanic families collected in gardens, parks, and median strips of Bogotá, were tested for phytoplasmas.

## 2. Materials and Methods

### 2.1. Plant Material

*Sambucus nigra* and *Sal. humboldtiana* surveys were conducted from September 2017 to March 2018, in urban zones in Bogotá (Timiza Park—4°36′29.5″ N 74°09′14.5″ W; Santa Helena Metropolitan Park—4°43′48.25″ N 74°3′2.14″ W; median strip 200 Street—4°46′58.3″ N 74°02′29.1″ W) and in rural areas in Cajicá (Campus Nueva Granada—4°56′31.6″ N 74°00′52.2″ W) and Funza (4°43′23.1″ N 74°11′41.1″ W). Of each tree species, 40 trees were selected at random (regardless of the presence or absence of symptoms), 10 in each location. They were used to determine the prevalence of symptoms associated with phytoplasmas diseases, that is the total number of symptomatic trees from the total number of sampled trees. Based on previous observations in other tree species [8,9], a list of possible symptoms was used to determine the health status of *Sa. nigra* and *Sal. humboldtiana*. The symptoms included crown yellowing, epicormic shoots, vertical branches, witches’ brooms, little leaves, decay, tufted foliage, loss of apical dominance, purple pigmentation of the leaves, dead branches, defoliation, atypic enlargement of apical branches, general deformation of the crowns, virescence, and phyllody. Additionally, low branches of 16 symptomatic *Sa. nigra* and 22 *Sal. humboldtinana* trees were cut and taken to the laboratory for DNA extractions. In the case of *Sal. humboldtiana*, additional leaf samples were collected in June 2018 from previously sampled trees. For the DNA extractions of the branch samples, secondary phloem was extracted by separating the bark from the sapwood. Additional symptom observations of both tree species were conducted in the Sabana de Bogotá in June 2023.

To study the role of weeds and grass as alternative hosts for phytoplasmas, samples were taken from locations at the north, south, east, west, and center of the city (4°35′56″57 N 74°04′51″30 W). In each zone, three 5 m^2^ quadrants were built in areas in which trees symptomatic for phytoplasmas were present: a park, a garden, and a median strip. In each quadrant, 10 weeds selected at random were sampled. At the moment of the collection, the presence of symptoms such as phyllody, virescence, leaf yellowing, abnormal shoot proliferation, and small leaves was recorded. To collect the insects associated with each weed, each plant was covered with a veil bag, pulled or cut from the soil and shaken into the bag. The insects were collected from the bag with a mouth aspirator and stored in 90% ethanol. In the laboratory, using morphological characters, the specimens were identified to order using a taxonomic key [19]. Each plant sample was divided in two. One part including leaves and stems was stored for DNA extraction. The other part was pressed, photographed, and mounted for taxonomic identification. In a different moment, samplings of the grass *Ce. clandestinus* were conducted in two parks of Bogotá, Botanic Garden of Bogota (BG) and “El Virrey” park (VP). Four symptomatic *F. uhdei*, *P. undulatum*, *M. grandiflora*, and *Q. humboldtii* trees were identified in each park. Grass samples were collected from a 5 m^2^ circumference area around each tree, and three grass leaf samples were collected per tree. In total, 48 grass samples were tested.

### 2.2. Plant DNA Extractions

Total DNA of 1 g of phloem bundles of *Sa. nigra* and *Sal. humboldtiana* trees was extracted by the modified chemical method of Prince et al. (1993) [20]. Total DNA extractions of weeds or grass samples, also starting from 1 g of leaf tissue, were obtained by the modified chemical method of Harrison et al. (2013) [21]. The DNA extracts were resuspended in TE buffer (10 mM Tris HCl, pH 8.0, 1 mM EDTA) and stored at −20 °C. To test for the presence of PCR inhibitors, the extracts were tested by PCR with the rpsF/rpsR2 primers, which anneal to an intron of the chloroplast rps16 gene, which is commonly present in plants [22]. If amplicons were obtained, it was assumed that the extracts did not contain PCR inhibitors. If no amplicons were obtained, the extracts were cleaned with the Monarch^®^ PCR & DNA Cleanup Kit^®^ (NEB, Ipswich, MA, USA) or the PowerClean^®^ DNA Clean-Up kit (MoBio, San Mateo, CA, USA), following the instructions of the manufacturers. The DNA extracts were stored at −20 °C.

### 2.3. Phytoplasma Detection and Sequence Analysis

Phytoplasmas were detected by nested PCR of the 16S rRNA gene using several combinations of universal primers. Usually, for the first PCR round, the P1A/P7A primers [23] were used, followed by primers R16F2n/R2, R16mF2/R16mR1 [24], or fU5/rU3 [25]. PCR reactions were conducted with 0.05 U of Taq polymerase (Bioline, London, UK), 1X PCR buffer, 1.5 mM MgCl_2_, 0.2 mM dNTPs (NEB, Ipswich, MA, USA), and 0.2 μM primers, in a final volume of 15 μL. Depending on the case, dilutions of total DNA extracts of the amplicons obtained in the first amplifications were diluted (1:20, 1:50, 1:100, or 1:200) and used as templates for nested reactions. The thermal profile of the PCR amplifications was initial denaturation (94 °C, 3 min), 35 denaturation cycles (94 °C, 45 s), annealing temperature (depending on the primer pair, 1 min), extension (70 °C, 1.5 min), and final extension (72 °C, 10 min). For primers P1A/P7 and fU5/rU3, the annealing temperature was 54 °C, and for R16F2n/R2 and R16mF2/R16mR1, 55 °C. Amplicons were analyzed by gel agarose electrophoresis at 1% in 1X TBE buffer TBE (89 mM Tris base, 89 mM boric acid, 2 mM EDTA), stained with ethidium bromide. Fragments of the 16SrRNA gene of MBS (Maize bushy stunt phytoplasma, group 16SrI-B) and ASHY (Ash yellows phytoplasma, group 16SrIVII-A) cloned in the pGEM^®^-T Easy Vector (Promega, Madison, WI, USA) were used as positive controls, and blank double-distilled water was used. To rule out cross-contamination in nested PCR reactions, the same blank tubes were used for the subsequent nested PCR reactions [11].

Positive amplicons obtained with the R16F2n/R16R2 primers in nested PCR reactions were analyzed by restriction fragment length polymorphism (RFLP) with the *Alu*I, *Hinf* I, *Hha*I, or *Mse*I (NEB, Ipswich, MA, USA) restriction enzymes, following the manufacturer’s instructions. The digested products were separated by 3% agarose gel electrophoresis in TBE 1X buffer as explained before. The RFLP patterns were compared with those obtained by Lee et al. (1998) [26] and with known MBA and ASHY controls.

Primers for the PCR amplification of the non-ribosomal *LeuS* gene were used to obtain phytoplasma sequences different from the 16S rRNA gene from grass samples. The Leufor1/Leurev1 followed by Leufor2/Leurev2 primers [27] were used in nested reactions. PCR conditions were as described for 16S rRNA gene except for the use of 2.5 mM MgCl_2_ and annealing temperature of 43 °C and 53 °C for the primary and nested PCR reactions, respectively. Positive control reactions were performed with genomic DNA of plants infected with Maize bushy stunt phytoplasma (MBS) or Ash yellows phytoplasma (ASHY) from North America maintained in vitro by grafting at the University of Bologna. The obtained amplicons were sequenced in Macrogen, Korea.

Selected amplicons were sequenced bidirectionally by Macrogen, Korea, with the same primers used in the PCR reaction. Each sequence was manually edited with Geneious Prime ver. 2024.0.5 (Biomatters, Auckland, New Zealand). Consensus sequences of each sample were built by the superposition of the forward and reverse sequences. The sequences were compared with the GenBank database using the basic local alignment search tool (BLASTn) algorithm [28], using standard parameters. Phytoplasma sequences obtained in this work were compared with 16S rDNA sequences of phytoplasmas of urban trees and Cicadellidae specimens of Bogotá that had been generated by us in previous works, and that were deposited in GenBank (https://www.ncbi.nlm.nih.gov/genbank/ (accessed on 3 July 2024 to 14 December 2024)). These sequences were aligned and used to build dendrograms that included representative sequences of phytoplasmas groups, using the software Geneious Prime ver. 2024.0.5 (Biomatters, Auckland, New Zealand), with the neighbor-joining algorithm and a value of 1000 bootstrap. The sequences obtained with primers R16F2n/R16R2 that were longer than 1250 pb were analyzed with the virtual tool *i*PhyClassifier ver.2024 [29].

## 3. Results

### 3.1. Symptom Observation

Systematic symptom observations of symptoms in *Sal. humboldtiana* and *S. nigra* were performed on 40 trees of each species, selected at random, in urban and rural locations of the Sabana de Bogotá. The prevalence of the disease was 100% for *Sal. humboldtiana* and 70% for *Sa. nigra* trees. All of the *Sal. humboldtiana* trees growing in both types of locations showed symptoms associated with phytoplasmas, including epicormic shootings and crown deformations; 88% had abnormal apical elongation of the branches and 67.5% abnormal elongation of branch internodes and branches growing in an upright abnormal position. Some trees had what appeared to be dead branches and small leaves. The observed symptoms were similar in both urban and rural locations (Figure 1, Table 1). However, recent observations of *Sal. humboldtiana* in the department of Cundinamarca and in the neighbor department of Boyacá show that the health status of the trees has deteriorated. In addition to apical dominance loss and abnormal internode elongation, the apical branches appear more defoliated and tufted foliage (branches with slow twig growth and short internodes that cause foliage to appear bunched) are present in every tree. Small leaves and yellowing are also widespread. Many trees have died since the initial observations.

In *S. nigra*, the symptoms were less clear, but the combination of several symptoms suggested phytoplasma disturbances in many of the observed trees. All the observed trees showed defoliation symptoms and 82.5% of them had dead branches. However, since these symptoms could be associated with many causes, trees were considered symptomatic for phytoplasmas if they displayed crown deformations, epicormic, atypic elongation of internodes, or tufted foliage in addition to defoliation or dead branches (Figure 1, Table 2). Therefore, on average, 82.5% of the observed trees were considered symptomatic for phytoplasmas. Leaf yellowing is a widespread symptom in these trees, but this symptom could be attributed to several causes. Symptoms in urban and rural locations were similar. The present symptoms in *Sa. nigra* appear to be similar to those observed years ago.

In urban areas, *Ce. clandestinus* is the dominant ornamental grass in parks, gardens, and median strips that surround the urban trees. Intermixed with grass, weeds are also widespread. In a survey conducted in 2016, 150 weed and grass samples were collected in 15 quadrants in different zones of Bogotá. The sampled plants belonged to 35 species Appendix A and the number of samples per species varied from 1 to 14. The sampled plants, including weeds and grass, were inspected in the field and later in the laboratory to determine possible symptoms associated with phytoplasmas. No evident symptoms were observed. Furthermore, insects of the Cicadellidae, Cixiidae, Psyllidae, Cercopidae, and Delphacidae families were not captured on the weeds using the described methodology, although flies, mosquitos, aphids, spiders, and beetles were readily collected in varying numbers.

### 3.2. Phytoplasma Detection

In total, 13 of 22 *Sal. humboldtiana* DNA extracts were positive by nested PCR using universal primers for phytoplasmas. For *Sa. nigra*, 13 of 16 samples tested positive for phytoplasmas. For both tree species, positive samples were obtained in urban and rural locations of the Sabana de Bogotá. RFLP or sequencing analysis of nested PCR amplicons showed the presence of group 16SrI in seven, group 16SrVII in two, and mixed infections of both phytoplasmas in two *Sal. humboldtiana* trees. In *Sa. nigra*, group 16SrI was detected in four and group 16SrVII in eight trees, and for one sample it was impossible to determine the phytoplasma species (Figure 2, Table 3).

Samples in which phytoplasmas were detected after two rounds of nested PCR with P1A/P7A, followed by R16mF2/R16mR1 and R16F2n/R16R2 primers. N = North, E = East, S = South, and W = West. BG = Botanic Garden of Bogotá, VG = “El Virrey” Park. Nested PCR tests were conducted on 150 weed samples from 35 species (1 to 14 samples depending on the species). These samples were collected in parks, gardens, and median strips in Bogotá Appendix A. For the primary PCR, P1A/P7A primers were used, followed by R16mF2/R16mR1 primers. However, the bands obtained for most of the samples were faint, so RFLP or sequencing analysis was not possible. Therefore, the positive amplicons were re-amplified with the R16F2n/R16R2 primers to produce enough DNA for RFLP or sequencing analysis. Positive plants belonged to the following species: *Amaranthus dubius* Mart. ex Thell., *Cymbalaria muralis* G. Gaertn., B. Mey. and Scherb, *Fumaria capreolata* L., *Gnaphalium cheiranthifolium* Mill., *Gnaphalium spicatum* Mill., *Lepidium bipinnatifidum* Desv., *Megathyrsus maximus* (Jacq.) B.K. Simon and S.W.L. Jacobs, *Myosotis sylvatica* Ehrh. ex Hoffm., *Plantago major* L., *Senecio vulgaris* L., *Sonchus oleraceus* L., *Spergula arvensis* L., and *Taraxacum officinale* F.H. Wigg. Samples of *Ce. clandestinus* were also infected. In total, 18 samples from 13 species were positive; 9 were identified as ‘*Ca*. P. asteris’ and 9 as ‘*Ca*. P. fraxini’ (Figure 2, Table 3). The complete list of the tested species is presented in the Appendix A.

Another survey was conducted to evaluate the infection of phytoplasmas in grass. *Ce. clandestinus* from two parks of Bogotá were tested for phytoplasmas. Grassland samples were collected within a radius of 5 m^2^ of symptomatic trees of *F. uhdei*, *P. undulatum*, *M. grandiflora,* and *Q. humboldtii* with symptoms of phytoplasma infection. In total, 32 of 48 samples were positive for phytoplasmas, 16 from the BG and 16 from the PV. ‘*Ca*. P. asteris’ and of ‘*Ca*. P. fraxini’ were detected by nested PCR followed by RFLP or sequencing analysis (Figure 2, Table 3). Eleven positive amplicons were analyzed by RFLP; two samples contained 16SrI group phytoplasmas, seven contained 16SrVII group phytoplasmas, and two had mixed infections of both phytoplasmas. Both groups of phytoplasmas were detected in the two parks (Figure 2, Table 3).

Amplicons of the 16S rRNA gene obtained by amplification of phytoplasmas of the tree, weed, and grass samples were analyzed by BLASTn. In all cases, the sequences belonged either to ‘*Ca*. P. asteris’ or ‘*Ca*. P. fraxini’ and the closest sequences corresponded to isolates from the Sabana de Bogotá, previously reported in NCBI. These sequences were used to construct trees by neighbor-joining, which included other sequences downloaded from GenBank. The sequences obtained in this work clustered with those of ‘*Ca*. P. asteris’ or ‘*Ca*. P. fraxini’; no other phytoplasmas species were detected in the samples studied (Figure 3).

### 3.3. Sequence Comparisons

The sequence identity of the 16S rRNA gene of trees, weeds, and grass was compared with sequences of phytoplasmas from plant and insect hosts previously reported in the Sabana de Bogotá. Figure 4 shows a simplified version of this analysis. The ‘*Ca*. P. asteris’ and ‘*Ca*. P. fraxini’ sequences grouped with sequences of the Sabana de Bogotá, with 100% of support in both cases. For instance, the sequence identity of ‘Ca. P. asteris’ from sample *Ce. clandestinus* 8 was high compared with that of trees such as *Croton* sp. 99%, *Q. humboldtii* 98.8%, *E. neomyrtifolia* 98.7%, *L. styraciflua* 99.4%, and *P. undulatum* 98.8%. It was also high compared with sequences obtained in this work from *Sa. nigra* 98.8% and *Sal. humboldtiana* 99.4% and weeds; *T. officinale* was 99.3%, *Gna. spicatum* 100%, *Am. dubius* 100%, and *Se. vulgaris* 100%. Furthermore, *Ca*. P. asteris’ sequences previously reported for Cicadellidae were also very similar such as in Typhlocibinae 99.7% and insect vectors *A. funzaensis* 99.8%, and *E. atratus* 98.9%.

The sequence identity of the ‘*Ca*. P. fraxini’ isolates obtained in this work were also similar to isolates reported for the Sabana de Bogotá. A comparison of the 16S rRNA phytoplasma sequence obtained from *Ce. clandestinus* (BG2) with sequences of trees previously reported from this zone shows sequence identities of 99.9% for *M. grandiflora* and 99.8% for *Q. humboldtii*, and for representative insect vectors *A. funzaensis* 99.2% and *E. atatus* 98.7%, and for a Typhlocibinae 98.3%. Compared with sequences obtained in this work, similarity with *S. nigra* was 99.9% and *Sal. humboldtiana* 99.9%, and with weeds *T. repens* 99.8%, *So. oleraceaus* 99.7%, *Gna. cherianthifolium* 99.9%, *Fu. capreolata* 99.8%, and *Lep. bipinnatifidum* 99.9%.

DNA extracts of *Ce. clandestinus* were tested by nested PCR using primers for the *LeuS* gene [27], in an attempt to amplify sequences of ‘*Ca*. P. asteris’ and ‘*Ca*. P. fraxini’. However, we were only able to amplify those of ‘*Ca*. P. asteris’. The sequences obtained from three isolates were used to construct a dendrogram including sequences of other species of phytoplasmas and sequences previously reported for trees of the Sabana de Bogotá. The sequence identities of four tree isolates Sabana de Bogotá varied between 93.8% and 100% compared with *Ce. clandestinus* 101 obtained in this work.

## 4. Discussion

As shown in previous works, our results suggest that the two main phytoplasma species found in trees, grass, and weeds of the Sabana de Bogotá are ‘*Ca*. P. asteris’ and ‘*Ca*. P. fraxini’ (Figure 3). Sequences of the 16S rRNA gene of these two phytoplasmas from a wide range of plant and insect (Cicadellidae) samples retrieved from GenBank were compared with sequences of *Sal. humboldtiana*, *Sa. nigra*, *Ce. Clandestinus,* and several weeds obtained in this work (Figure 4). In the Sabana de Bogotá, there is evidence of phytoplasmas of both groups infecting Cicadellidae, so we included in our analysis some of the sequences reported previously in GenBank [11,12,18]. For both phytoplasmas, the percentage of sequence identity compared among many different plant and insect hosts was over 98.7%, suggesting that in both cases, closely related strains circulate in the environment, infecting species in a large number of botanical families and Cicadellidae species. The distribution patterns of phytoplasmas in the ecosystems depend on the susceptibility of plants to phytoplasmas and on the feeding habits of the infected insect vectors [3].

‘*Ca*. P. asteris’ and ‘*Ca*. P. fraxini’ have been previously reported in the Sabana de Bogotá infecting nine tree species growing in urban and semiurban areas [6,7,8,9,10,11], as well as potato and strawberry crops [7,12]. This work is an attempt to describe the complex epidemiological relationships that explain the wide distribution of these phytoplasmas in the ecosystem of the Sabana de Bogotá and the implications for its future management. The particular characteristics of phytoplasma diseases require, for their management, integrated approaches combining components of cultural, physical, biological, resistance, and chemical control [30], which, in turn, require knowledge of the epidemiological variables involved.

In this work, ‘*Ca*. P. asteris’ and ‘*Ca*. P. fraxini’ were detected in *Sal. humboldtiana* where the most notorious symptoms included epicormic shoots, crown deformations, and abnormal elongation of apical shoots (Table 1), in both urban and rural locations. The prevalence of the disease was 100%, but as explained before, the health status of these trees has sharply declined in the past years making the symptoms more visible. Burckhardt and Pinzón (2024) [31] reported the recent introduction of the North American psyllid species *Bactericera minuta* Crawford 1910 (Hemiptera: Psylloideae) in *Sal. humboldtiana* trees in the Colombian department of Cundinamarca, where the Sabana de Bogotá is located, and in the neighboring department of Boyacá. Mild phytoplasma symptoms had been observed in *Sal. humboldtiana* since 2005 in the Sabana de Bogotá (L. Franco-Lara, personal communication) but the worsening of the health status of the trees could be due to the stress imposed by *B. minuta*. It seems that *B. minuta* associates exclusively with *Sal. humboldtiana* and is widely distributed in Cundinamarca and Boyacá [31]. Psyllid species belonging to the genus *Cacopsylla* (Hemiptera: Psylloidea) are implicated in the transmission of phytoplasmas in Europe, to fruit trees such as pear, apple, stone fruit, and peach [32], and of the genus *Diaphorina* to citrus [33]. The role of *B. minuta* in the transmission of phytoplasmas among *S. humboldtiana* trees is unknown. Phytoplasma infections have been reported for some species of the genus *Salix* including *Salix babylonica, Salix alba*, and *Salix tetradenia* [34,35,36], but so far this is the first report for *Sal. humboldtiana*. ‘*Ca*. P. asteris’ and ‘*Ca*. P. fraxini’ were also detected in *Sa. nigra*. In this case, phytoplasmas symptoms were less evident than in *Sal. humboldtiana*. Defoliation, dead branches, unspecific crown deformations, and epicormic shoots were present in 82.5% or more of the surveyed *Sa. nigra* trees (Table 2). Although yellowing is a characteristic symptom of phytoplasma diseases, it can be associated with other causes, so in this work we did not record its presence. However, additional observations show that indeed leaf yellowing is a very common symptom in these trees in the Sabana de Bogotá and in other areas in Colombia, usually accompanied by the symptoms described above (L. Franco-Lara, personal communication). Phytoplasma infections in *Sa. nigra* has been reported before [37].

In our tests, there were nine samples of *Sal. Humboldtiana* and three of *Sa. nigra* that were clearly symptomatic but in which phytoplasmas were not detected. This could be attributed to several causes. The heterogeneous distribution of phytoplasmas in plant tissues has been observed before [38,39,40] (unpublished results). Since in these trees there are limitations to accessing the upper part of the crowns, the samplings were limited to the lower branches of the trees. The distribution of phytoplasmas inside the plants is unknown, therefore, it is possible that in some cases we sampled branches with undetectable concentrations of phytoplasmas. This may have had less impact on *Sa. nigra* because these are smaller trees. Another possibility is that these DNA extracts may have had phenolic substances that inhibit PCR as has been noted before for woody plants [40,41].

We report for the first time phytoplasma infection in weeds growing near infected trees in the Sabana de Bogotá including *Am. dubius*, *Cy. muralis*, *Fu. capreolata*, *Gna. cheiranthifolium*, *Gna. spicatum*, *Lep. bipinnatifidum*, *Me. maximus*, *My. sylvatica*, *Pl. major*, *Se. vulgaris*, *So. oleraceus*, *Sp. arvensis*, and *T. officinale* (Table 3). In infected plants of these species, stand out symptoms such as yellowing, deviation from normal morphology, small leaves, phyllody, virescence, reddening, etc., were not observed. Phytoplasmas asymptomatic infections have been reported before [42,43]. In Poland, ‘*Ca*. P. asteris’ was detected infecting several weed species, but although they were asymptomatic, they might play a role as inoculum source of phytoplasmas for *Brassica napus* crops [44]. The detection of phytoplasmas in weeds has been reported before [45]; furthermore, phytoplasmas have been previously reported infecting plants of genera *Amaranthus*, *Lepidium*, *Sonchus*, *Plantago*, *Senecio*, *Sonchus*, and *Taraxacum* [46,47]. Our attempts to sample possible Cicadellidae or Psyllidae insect specimens from the weeds failed, although the method allowed for the capture of other small arthropods from them. In many pathosystems, the epidemiology of phytoplasma diseases is associated with the presence of susceptible weeds and insect vectors. For instance, in Italy and Germany the vector *Hyalesthes obsoletus* (Hemiptera: Cixiidae), among others, seems to be implicated in the transmission of the Stolbur phytoplasma from the weeds *Convolvulus arvensis* and *Urtica dioica* to grapevines [48,49]. In India, the presence of phytoplasmas of group 16SrII in alternative hosts has been associated with diseases in crops such as sesame, cow pea, red bean, bamboo, and papaya [50]. In Saudi Arabia, the weeds *Chenopodium morale*, *Plantago lanceolata*, and *Convolvulus arvensis*, as well as the leafhopper *Empoasca decipiens* (Hemiptera: Cicadellidae), have been implicated in the dispersal of phytoplasmas of group 16SrII to lime orchards [51].

Our results show that *Ce. clandestinus* is an asymptomatic host of ‘*Ca*. P. asteris’ and ‘*Ca*. P. fraxini’ (Table 3). Previous indirect observations suggested that this grass was infected with phytoplasmas because transmission essays using insect vectors *A. funzaensis* and *E. atratus* collected from grass growing near infected trees, allowed for the experimental transmission of both species of phytoplasmas to red bean and potato plants [12,18]. Indirect evidence of the presence of phytoplasmas in *Ce. clandestinus* was obtained by nested PCR amplification of the 16S rRNA gene and further confirmation by RFLP and sequence analysis. The amplification of the *LeuS* gene of ‘*Ca*. P. asteris is further evidence of phytoplasma infection in *Ce. clandestinus* (Figure 5). Attempts to amplify the *LeuS* gene of ‘*Ca*. P. fraxini’ were unsuccessful. *Ce. clandestinus* samples were taken from different locations of the Sabana de Bogotá in different years, suggesting that phytoplasma amplification was not an incidental observation (Table 3). In many parts of the world, grasses are known hosts of phytoplasmas and play a key role as source of inoculum for the dispersion of these pathogens [52,53,54].

Our observations show that *Ce. clandestinus* is a key player in the epidemiology of the phytoplasma diseases of the Sabana de Bogotá since it is a host for the phytoplasma pathogens and for the insect vectors. There are about 30 species of Cicadellidae that inhabit *Ce. clandestinus* grasslands [16,17]. Nymphs of all developmental stages of *A. funzaensis* and *E. atratus* and of other Cicadellidae have been collected on *Ce. Clandestinus,* which suggests that it is a reproductive host for them (L. Franco-Lara, unpublished results). Moreover, it has been suggested that *A. funzaensis* and *E. atratus* are polyphagous since they have also been captured on *Q. humboldtii* and *S. tuberosum* [11,12] and have been experimentally fed on red beans and celery [18]. *Ce. clandestinus* is an introduced grass species, now naturalized in Colombia. In the Sabana de Bogotá, it thrives in urban and rural environments and has displaced the native Colombian grasses in many parts of Colombia over 1500 m of altitude [15]. Therefore, a possible scenario is that polyphagous insect vectors live and reproduce in the grass, which grows extensively in the area and occasionally visit other plants including trees and crops. In this process, they disperse both species of phytoplasmas, which, in turn, successfully infect many species of botanic species, as has been suggested before [55]. Since it is so widespread and well adapted to the conditions, grass is in practice impossible to remove, which hampers management strategies for this disease. Moreover, Colombia is a tropical country and there are not meteorological seasons. The fact that the temperature is relatively constant, even in the rainy periods, may explain in part the ample dispersion of phytoplasmas in this area of Colombia. In these conditions, woody and herbaceous plants are always green, allowing phytoplasmas survival and making them a permanent source of inoculum. Insect vectors are also present throughout the year as we have reported before [17]. These conditions allow for the presence of hosts, pathogens, and insect vectors year-round, in permissive climatic conditions, which may account for the ample dispersion of the disease. The sum of all these conditions poses a complicated epidemiological problem and the need for a different approach to tackle this disease.

## Figures and Tables

**Figure 1 microorganisms-13-00967-f001:**
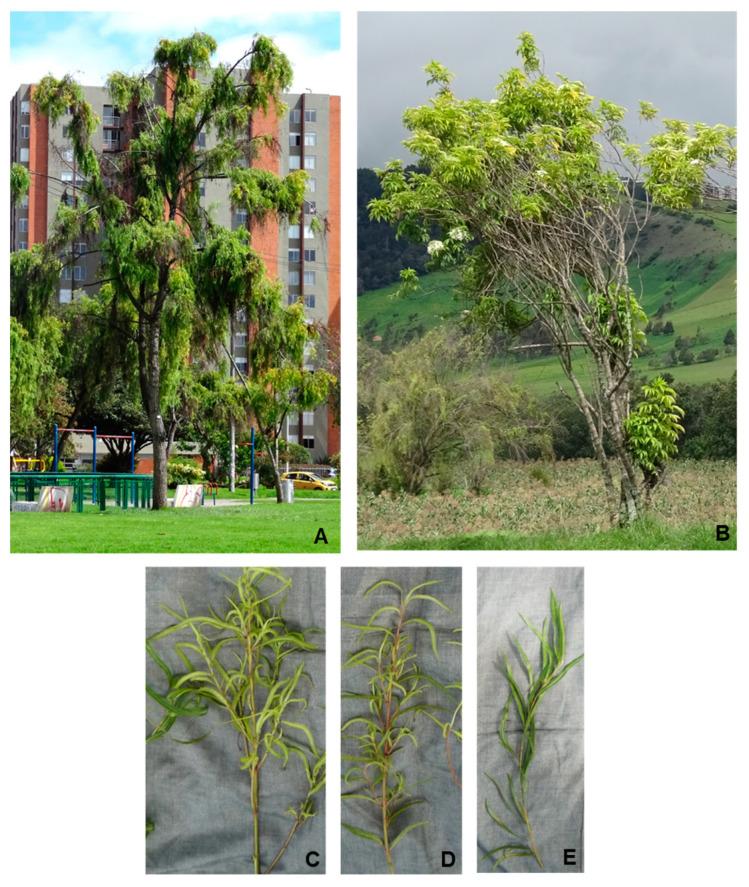
(**A**) *Salix humboldtiana* tree showing symptoms such as dead branches, tufted foliage, defoliation, abnormal elongation of apical shoots, bunches of dead leaves, and changes in the crown architecture. (**B**) *Sambucus nigra* tree showing symptoms such as yellowing, dead branches, epicormic shoots, tufted foliage, and crown deformation. (**C**,**D**) *Salix humboldtiana* secondary branches with small, yellow leaves and abnormal leaf pattern distribution. (**E**) *Salix humboldtiana* secondary branch with leaves showing a normal growth pattern.

**Figure 2 microorganisms-13-00967-f002:**
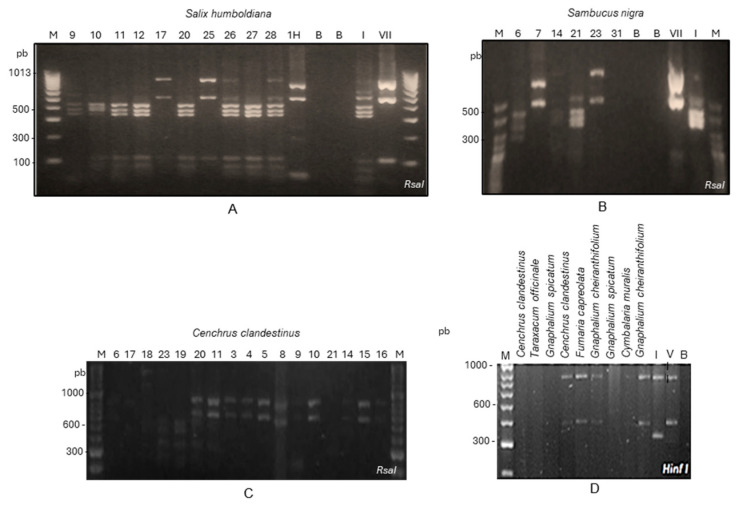
RFLP analysis of amplicons obtained by nested PCR using primers P1A/P7A followed by R16F2n/R2 of (**A**) *Salix humboldiana*, (**B**) *Sambucus nigra,* and (**C**) *Cenchrus clandestinus* samples, digested with *Rsa*I. (**D**) Amplicons obtained with P1A/P7A, R16mF2/R16mR1, and R16F2n/R2 of weed and grass samples, digested with *Hinf* I. M, molecular weight marker (**A**) 100 bp (Bioline), (**B**) 50 bp (Bioline), (**C**) 100 bp (Bioline). Numbers indicate sample numbers. Positive controls were I (MBS phytoplasma group 16SrI) and VII (Ash yellows phytoplasma, group16SrVII). B (blank) indicates water control.

**Figure 3 microorganisms-13-00967-f003:**
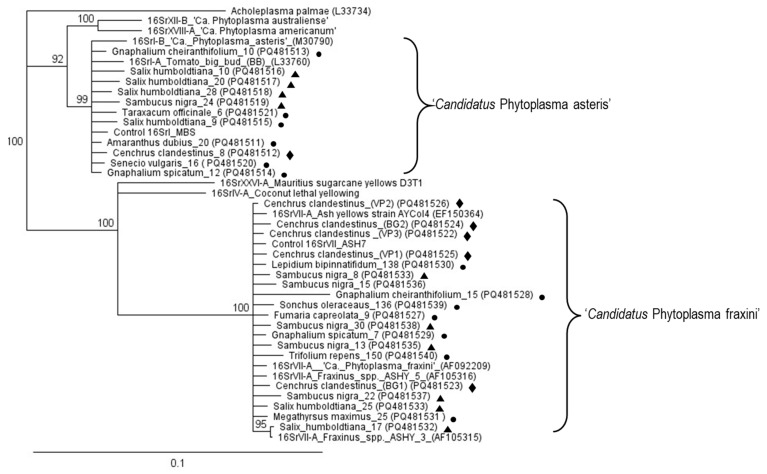
Dendrogram constructed with phytoplasmas sequences of the 16S rRNA gene obtained from trees (triangles), weeds (circles), and grass (rhombus) samples obtained in this work, and selected sequences from GenBank. The names given to the sequences correspond to the plant hosts in which they were detected. The tree was constructed using the neighbor-joining method with 1000 bootstrap replicates. *Acholeplasma palmae* was used as an outgroup. GenBank accession numbers are shown in parentheses.

**Figure 4 microorganisms-13-00967-f004:**
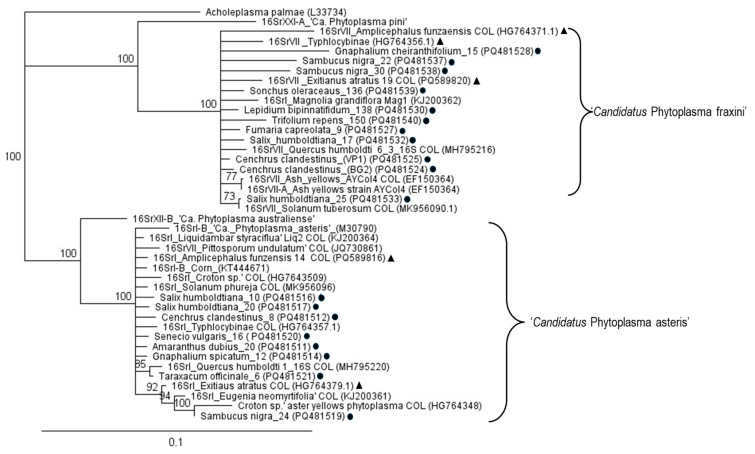
Dendrogram constructed with phytoplasmas sequences obtained of the 16S rRNA gene from trees, weeds, and grass obtained in this work (circles), and including sequences from other plant and insect hosts (triangles) from the Sabana de Bogotá. The names given to the sequences correspond to the hosts in which they were detected. The sequences that correspond to Colombian isolates are labelled as COL. The tree was constructed using the neighbor-joining method with 1000 bootstrap replicates*. Acholeplasma palmae* was used as an outgroup. GenBank accession numbers are shown in parentheses.

**Figure 5 microorganisms-13-00967-f005:**
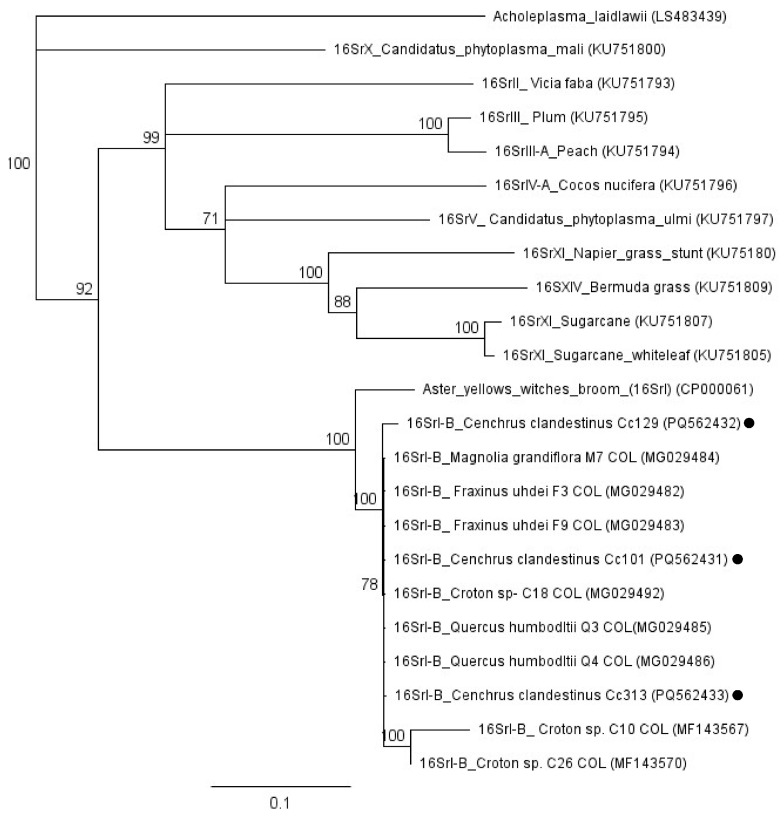
Dendrogram constructed with *LeuS* gene sequences of phytoplasmas and containing sequences obtained from *Cenchrus clandestinus* in this work (circles) and from isolates previously reported from other plant hosts of the Sabana de Bogotá. The tree was constructed using the neighbor-joining method with 1000 bootstrap replicates. *Acholeplasma laidlawii* was used as an outgroup. GenBank accession numbers are shown in parentheses.

**Table 1 microorganisms-13-00967-t001:** Symptom prevalence on urban and rural *Salix humboldtiana* trees in the Sabana de Bogotá (2017–1018).

Symptoms	% Prevalence (n = 40)				
	Timiza ^1^	Median Strip 200 St ^1^	Funza ^2^	Cajicá ^2^	Average
Epicormic shoots	100	100	100	100	100
Crown deformation	100	100	100	100	100
Abnormal elongation of apical shoots	100	80	80	90	87.5
Vertical branches	70	60	60	80	67.5
Atypical elongation of internodes	60	80	70	60	67.5
Dead branches	10	40	30	0	20
Small leaves	0	0	0	10	2.5

^1^ Urban location, ^2^ rural location.

**Table 2 microorganisms-13-00967-t002:** Symptom prevalence on urban and rural *Sambucus nigra* trees in the Sabana de Bogotá (2017–1018).

Symptoms	% Prevalence (n = 40)				
	Timiza 1	Santa Helena 1	Funza 2	Cajicá 2	Average
Defoliation	100	100	100	100	100
Dead branches	80	80	80	90	82.5
Crown deformation	70	70	90	100	82.5
Epicormic shoots	70	60	80	80	72.5
Vertical branches	10	40	30	40	30
Abnormal elongation of apical shoots	20	30	20	30	25
Tufted foliage	30	0	20	10	15
Small leaves	0	0	0	10	2.5

^1^ Urban location, ^2^ rural location.

**Table 3 microorganisms-13-00967-t003:** Detection of phytoplasmas by nested PCR and classification by RFLP and sequencing analysis of trees, weeds, and grass samples.

Species	Locations	Positive/Total Samples	Primers P1A/P7A R16mF2/R16mR1	Primers P1A/P7A R16F2n/ R16R2	Primers P1A7P7 AfU5/rU3	RFLP			Sequencing	
						16SrI	16SrVII	16SrI + 16SrVII	16SrI	16SrVII
** *Salix humboldtiana* **	Timiza Park	1/5	1				1			
	200 St	4/5	4			4			2	
	Cajicá	3/6	2	1		1	1		1	1
	Funza	5/6	5			2	2	1	1	1
** *Sambucus nigra* **	Timiza Park	3/4	2	2	1	1	1			1
	Santa Helena Park	3/4	1	2		1				2
	Cajicá	4/4	2		2	1	1		1	1
	Funza	3/4	1		2					2
** *Amaranthus dubius* **	Bogotá—N	1/11		1 *					1	
** *Cymbalaria muralis* **	Bogotá—E	1/5		1 *					1	
** *Fumaria capreolata* **	Bogotá—E	1/3		1 *						1
** *Gnaphalium cheiranthifolium* **	Bogotá—E	2/3		2 *					1	1
** *Gnaphalium spicatum* **	Bogotá—E, N	2/5		2 *					1	1
** *Lepidium bipinnatifidum* **	Bogotá—W, N	2/7		2 *		1				1
** *Megathyrsus maximus* **	Bogotá—E	1/2		1 *						1
** *Myosotis sylvatica* **	Bogotá—E	1/2		1 *					1	
** *Plantago major* **	Bogotá—S	1/3		1 *			1			
** *Senecio vulgaris* **	Bogotá—W	2/10		2 *			1			1
** *Sonchus oleraceus* **	Bogotá—N	1/5		1 *						1
** *Spergula arvensis* **	Bogotá—E	1/1		1*					1	
** *Taraxacum officinale* **	Bogotá—E, N, S	2/14		2 *		1			1	
** *Cenchrus clandestinus* **	Bogotá—E	2/6		2 *		1				1
	Bogotá—BG	16/24		16		1	3	1		2
	Bogotá-—PV	16/24		16		1	4	1		4

Note that for the trees, there is not a strict coincidence in the number of samples in which the phytoplasmas species were detected by RFLP and sequencing analysis, since sometimes it was possible to deduce the species by one method but not the other. Asterisks indicate that in these samples two nested PCR tests were subsequently conducted after the primary PCR with the P1A/P7A primers, first with the R16mF2/R16mR1 primers and finally with the R16F2n/ R16R2 primers.

## Data Availability

Sequences produced in this work were submitted and published in GenBank. Accession numbers are given in the corresponding dendrograms in Figure 3, Figure 4 and Figure 5.

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
