# Peer review of "The Role of Grass in the Epidemiology of a Phytoplasma Disease Affecting Trees and Other Plants of the Sabana de Bogotá, Colombia"

_microorganisms, 2025, doi:10.3390/microorganisms13050967_

Round 1
Reviewer 1 Report
Comments and Suggestions for Authors
Review Report
The manuscript presents a significant and timely study on the distribution and epidemiology of ‘Candidatus Phytoplasma asteris’ and ‘Candidatus Phytoplasma fraxini’ in the Sabana de Bogotá. The authors provide compelling evidence for the presence of these phytoplasmas across a wide range of host plants, including trees, grasses, and weeds, as well as in insect vectors from the Cicadellidae family. This research offers valuable insights into the complex interactions between phytoplasmas, their plant hosts, and insect vectors, with important implications for disease management in this agriculturally important region.
The study is generally well-organized and presents meaningful findings. However, several areas could benefit from further refinement to improve clarity, methodological transparency, and the overall impact of the work. Addressing these points will strengthen the manuscript and enhance its contribution to the field of phytoplasma epidemiology.
Comments for Authors
- Sampling took place over a broad time span (September 2017 to June 2023), but the temporal distribution of sample collections (frequency, seasonality, consistency) is unclear. Phytoplasma incidence can be seasonal; this variability should be acknowledged and accounted for in the methods.
- Insect collection is briefly described, but there’s a lack of detail regarding identification procedures. It’s unclear whether morphological or molecular techniques were used to identify Cicadellidae vectors, and no mention is made of taxonomic references or keys used.
- There is no mention of how prevalence data for phytoplasmas (from trees, weeds, grasses, and insects) were statistically analyzed or if prevalence rates were compared across host species, regions, or years.
Author Response
- Sampling took place over a broad time span (September 2017 to June 2023), but the temporal distribution of sample collections (frequency, seasonality, consistency) is unclear. Phytoplasma incidence can be seasonal; this variability should be acknowledged and accounted for in the methods.
Answer: Colombia is located in the tropical zone, where there are no metereological seasons. For instance, in Bogotá, the average daily temperature is 18°C and the night temperature is 9°C all the year, as can be seen in the following link: https://www.meteoblue.com/en/weather/historyclimate/climatemodelled/bogot%C3%A1_colombia_3688689)
In our experience, phytoplasmas can be consistently detected by nested PCR in infected plants throughout the year. In other works made by our group we have been able to detect ‘Candidatus Phytoplasma asteris’ and ‘Candidatus Phytoplasma fraxini’ in any month of the year. For example, in a PhD thesis the student tested 238 Andean oak tree samples in Bogotá, from May to August 2017. The trees were selected randomly, and 94% were positive for phytoplasmas (Lamilla et al., 2022). At least 30 of those trees were resampled in the next year, from January to June or September to December, and phytoplasmas were readily detected in all. Similarly, in potato crops of Solanum tuberosum and Solanum phureja in the Cundinamarca state, where Bogotá is located, ‘Ca. P. asteris’ and ‘Ca. P. fraxini’ were detected in June to October 2015 and April to October 2016 (Franco-Lara et al., 2023).
The fact that the temperature is relatively constant, even in the rainy periods, may partly explain the ample dispersion of phytoplasmas in this area of Colombia. In these conditions, woody and herbaceous plants are always green, allowing phytoplasmas survival and making a permanent source of inoculum. Insect vectors are also present throughout the year, as we observed (Silva-Castaño et al., 2020). There are at least two insect vectors known in the area both of them are polyphagous, which favor phytoplasma dispersion.
In the present work, phytoplasmas were detected in trees, grass and weeds in different years and months. This suggests that the disease has been in a stable condition through time and is not an sporadic phenomenon. A small paragraph was introduced in the discussion to clarify this observation.
- Insect collection is briefly described, but there’s a lack of detail regarding identification procedures. It’s unclear whether morphological or molecular techniques were used to identify Cicadellidae vectors, and no mention is made of taxonomic references or keys used.
Answer: In this work, the sampling effort was on plant specimens. However, we unsuccesfully attempted to collect Cicadellidae, Cixiidae, Psyllidae, Cercopidae of Delphacidae from weeds (excluding grass). Flies, mosquitos, aphids, spiders and beetles were readily collected in varying numbers on the weeds. To determine if these specimens belonged to the groups of interest, they were indentified to order using the taxonomic key of Tripplehorn, C. A., & Johnson, N. F. (2005). Borror and DeLong’s introduction to the study of insects. Thomson Brooks/Cole, Belmont, California. This reference was included in the paper. No further identification was performed because it was irrelevant for this work.
In previous works by our group in which we studied possible insect vectors, we collected Cicadellidae and Pyslloidea on grass (Silva –Castaño et al., 2020; Silva-Castaño et al., 2023; unpublished results), trees of several species (Lamilla et al., 2022; Silva-Castaño et al., 2024; unpublished results) and potatoes (unpublished results). These species were identified using external morphological characteristics using taxonomic keys (Dietrich, 2005; Fernández & Sharkey, 2006; Krishnankutty et al., 2016; Linnavuori, 1959; Young, 1977; Zahniser & Dietrich, 2008). The initial taxonomic identifications were performed with the support of Dr Michael R Wilson, who trained us in the taxonomic identification of Hemiptera. Later, we developed DNA barcodes for several Cicadellidae species which have been deposited in the BOLD (Barcode of Life Data Systems) (https://v3.boldsystems.org/). Most of the Colombian Cicadellidae and Psyllidae species are new to science, and most if not all of the sequences uploaded into public databases have been generated by us. In this work, we compared the 16S rDNA phytoplasma sequences of several Cicadellidae specimens that we had collected, identified taxonomically, used to generate DNA barcodes, tested for phytoplasma infection and included in the public databases. Since these sequences were not produced in this work, we did not describe their identification process in the paper. Furthermore, we also used sequences of other infected trees that produced by us in previous projects. A paragraph was included in the materials and methods to clarify this point.
- There is no mention of how prevalence data for phytoplasmas (from trees, weeds, grasses, and insects) were statistically analyzed or if prevalence rates were compared across host species, regions, or years.
Answer: The objective of this work was to provide evidence of the occurrence and analyse the epidemiological implications of the presence of ‘Ca. P. asteris’ and ‘Ca. P. fraxini’ in trees, grass, and weeds in the Sabana de Bogotá, not to estimate the prevalence of the disease or the pathogen and therefore, the experimental design does not allow us to produce this information. There are several considerations in this regard. In the past, we have estimated the symptom and the phytoplasma prevalence for nine tree species, and it varied from 33% to 100% (Franco-Lara et al., 2014). Trees are long-lived plants, and they don´t recover from the disease, but the insect vectors and phytoplasma inoculum is permanent, so the most likely result is that the prevalence will increase with time. Grass is also a long-lived species, but its growth pattern in culms does not allow to estimate the number of plants in a grassland. In this case again, the likely tendency is prevalence increase. Observations not included in the paper show that in many áreas of the Sabana de Bogotá the grass is infected with phytoplasmas, and as said before, we predict that it will further increase. It would be interesting to measure the phytoplasma dispersionn speed, but we haven´t done it.
References
Dietrich, C. H. (2005). Keys to the families of Cicadomorpha and subfamilies and tribes of Cicadellidae (Hemiptera: Auchenorrhyncha). Florida Entomologist, 88, 502–517.
Fernández, F., & Sharkey, M. J. (2006). Introducción a los Hymenoptera de la Región Neotropical (No. LC-0224). Sociedad Colombiana de Entomología, SOCOLEN Universidad Nacional de Colombia.
Franco-Lara L, Perilla-Henao LM. Phytoplasma diseases in trees of Bogotá, Colombia: a serious risk for urban trees and Crops. In A. Bertaccini (ed.): Phytoplasmas and phytoplasma disease management: how to reduce their economic impact, Bologna, Italy; 481 2014; pp. 90-100.
Franco-Lara, L., Varela-Correa, C. A., Guerrero-Carranza, G. P., & Quintero-Vargas, J. C. (2023). Association of phytoplasmas with a new disease of potato crops in Cundinamarca, Colombia. Crop Protection, 163, 106123.
Krishnankutty, S. M., Dietrich, C. H., Dai, W., & Siddappaji, M. H. (2016). Phylogeny and historical biogeography of leafhopper subfamily Iassinae (Hemiptera: Cicadellidae) with a revised tribal classification based on morphological and molecular data. Systematic Entomology, 41(3), 580–595.
Lamilla, J., Solano, C. J., & Franco‐Lara, L. (2022). Epidemiological characterization of a disease associated with phytoplasmas in Andean oak, Quercus humboldtii Bonpland, in Bogotá—Colombia. Forest Pathology, 52(2), e12730.
Linnavuori, R. (1959). Revision of the Neotropical Deltocephalinae and some related Subfamilies (Homoptera). Annales Zoologici Societatis Vanamo, 21(1), 1–370.
Silva-Castaño, A. F., Brochero, H., & Franco-Lara, L. (2024). Insects as potential vectors of phytoplasmas in urban trees in a mega-city: a case study in Bogotá, Colombia. Urban Ecosystems, 27(5), 1509-1525.
Silva-Castaño, A. F., Franco-Lara, L., & Brochero, H. (2023). Exitianus atratus (Hemiptera: Cicadellidae) a vector of phytoplasmas: morphometry, DNA barcode and phylogenetic relationships for nymphs and adults from Colombia. International Journal of Tropical Insect Science, 43(2), 495-506.
Silva-Castaño, A. F., Wilson, M. R., Brochero, H. L., & Franco-Lara, L. (2020). Biodiversity, bugs, and barcodes: the Cicadellidae associated with grassland and phytoplasmas in the Sabana de Bogotá, Colombia. Florida Entomologist, 102(4), 755-762.
Young, D. (1977). Taxonomic study of the Cicadellinae (Homoptera: Cicadellidae). 2. New world Cicadellini and the genus Cicadella. Technical Bulletin-North Carolina Agricultural Experiment Station (USA), 239, 1–1135.
Zahniser, J. N., & Dietrich, C. H. (2008). Phylogeny of the leafhopper subfamily Deltocephalinae (Insecta: Auchenorrhyncha: Cicadellidae) and related subfamilies based on morphology. Systematics and Biodiversity, 6(1), 1–24.
Reviewer 2 Report
Comments and Suggestions for Authors
The manuscript microorganisms-3553787 describes the detection of phytoplasmas in various plants in one of the regions (Bogota) of Colombia. The authors presented the results of a large work; however (in my opinion) not all the necessary aspects are discussed in the manuscript.
The authors assessed the phytoplasma disease visually and using PCR tests. However, the data do not match. For trees, the PCR method detected fewer cases of infection than were observed during inspection. For weeds, on the contrary, the plants were asymptomatic, but 18 samples were positive for PCR analysis. These discrepancies in the data are not discussed or explained in any way.
The authors make assumptions about the ways of maintaining and spreading the disease caused by phytoplasmas. And insect carriers play one of the main roles in this. However, the authors were unable to detect phytoplasmas in insects because they did not find the necessary species during collection. In my opinion, this aspect is also a weak point of the manuscript, which the authors do not discuss and explain properly.
Minor remarks:
Lines 2 and 28: Why do the authors use the term "epidemiology"? The manuscript says nothing about human diseases caused by phytoplasmas?
Line 2: Since the manuscript does not show the involvement of insects, how appropriate is it to use "Insect" in the title of the manuscript? I find the title of the manuscript incorrect.
Lines 91-93: "40 trees were selected at random" - please clarify this phrase. Were the same number of trees selected in each location? Was the presence of signs of disease caused by phytoplasmas already taken into account during the selection? What is the size of the populations of the studied plants in the selected locations? That is, how representative are the 40 trees for the studied populations?
Line 182 and following: The authors use "S." to denote different genera of plants: Salix, Sambucus, Senecio, Sonchus. This is very inconvenient for perceiving the manuscript information. Such a remark for "L.": both Liquidambar and Lipidium; for "T.": both Trifolium and Taraxacum.
Lines 185 and 379: "70%" - this value does not correspond to the data in Table 2. And if only trees with symptoms of the disease were selected, why did not 100% of the trees have signs of the disease.
Tables 1 and 2: Specify the number of trees from each location.
Lines 235-236: Forty trees of each species were analyzed, but 22 Salix humboldtiana and 16 Sambucus nigra samples were PCR detected for phytoplasmas. Why? How were samples selected for PCR analysis?
Table 3. What does "Especies" mean?
Lines 257-284 (discussion of Table 3): For the prevention of diseases caused by phytoplasmas, it is interesting to discuss the plants in which phytoplasmas were not detected.
Author Response
The manuscript microorganisms-3553787 describes the detection of phytoplasmas in various plants in one of the regions (Bogota) of Colombia. The authors presented the results of a large work; however (in my opinion) not all the necessary aspects are discussed in the manuscript.
The authors assessed the phytoplasma disease visually and using PCR tests. However, the data do not match. For trees, the PCR method detected fewer cases of infection than were observed during inspection. For weeds, on the contrary, the plants were asymptomatic, but 18 samples were positive for PCR analysis. These discrepancies in the data are not discussed or explained in any way.
Answer: A sentence was introduced in materials and methods correcting the number of tree samples analysed by PCR for each species. Of 40 S. humboldtiana 22 were clearly symptomatic, and of those 13 were positive for phytoplasmas by nested PCR. Of 40 surveyed S. nigra trees, 16 had clear symptoms of phytoplasmosis. Of those trees, 13 were positive for phytoplasmas. This means that for S. humboldtiana there were 9 samples and and for S. nigra 3 samples that were symptomatic but in which phytoplasmas were not detected. In a study conducted by our group, we studied the distribution of ‘Candidatus Phytoplasma asteris’ and ‘Candidatus phytoplasma fraxini’ in the crown of 12 Quercus humboldtii trees. In each tree, we sampled the upper right and left branches, the middle left and right branches and the lower left and right branches and tested for phytoplasmas by qPCR. We couldn´t find any reproducible pattern to explain the distribution or concentration of phytoplasmas in the crown of these trees. Our results show that these two phytoplasmas were randomly distributed in the tree, and therefore there were branches in which phytoplasmas were not detected even if the tree was infected (unpublished results). The heterogeneous distribution within plants has been reported before (Christensen et al., 2004; Siddique et al., 1998 Herath et al., 2010).
However, when working with trees, a limitation is the access to the upper part of the crown. In the case of Sa. humboltiana, the large tree size makes it almost impossible to sample the upper or middle branches of the tree. Additionally, frequently, it is also impossible to access more than one branch making the sample small to the tree size. This may have had less impact on Sa. nigra because these are smaller trees. Another possibility is that these DNA extracts may have had phenolic substances that inhibit PCR as has been noted before for woody plants (Osman et al., 2006; Herath et al., 2010). A paragraph was introduced in the discussion to address this point.
The authors make assumptions about the ways of maintaining and spreading the disease caused by phytoplasmas. And insect carriers play one of the main roles in this. However, the authors were unable to detect phytoplasmas in insects because they did not find the necessary species during collection. In my opinion, this aspect is also a weak point of the manuscript, which the authors do not discuss and explain properly.
In this work emphasis was on the presence of phytoplasmas infecting many botanical families. However, we also provide indirect evidence of phytoplasma infection in Cicadellidae from the area, including two insect vectors Amplicephalus funzaensis and Exitianus atratus, using sequences reported by us in GenBank. In previously published and unpublished research, we have plenty of evidence of the large numbers of infected Cicadellidae in the Sabana de Bogotá, but this is was not mentioned in the paper to avoid excessive self-citing. However, an explanation was provided in the discussion to make the point clear.
Minor remarks:
Lines 2 and 28: Why do the authors use the term "epidemiology"? The manuscript says nothing about human diseases caused by phytoplasmas?
In the Merriam-Webster dictionary, epidemiology is defined as a “Branch of the medical science that deals with the incidence, distribution and control of a disease in a population” or “the sum of the factors controlling the presence of absence of a disease or pathogen”. In this work, we abide by the second definition, as many papers in phytopathology do. These are some examples: i) Jeger, M. J. (2020). The epidemiology of plant virus disease: Towards a new synthesis. Plants, 9(12), 1768. ii) Trebicki, P. (2020). Climate change and plant virus epidemiology. Virus research, 286, 198059. iii) Wang, H. (2023). Epidemiology and control of fungal diseases in crop plants. Agronomy, 13(9), 2327. iv) Pangga, I. B., Macasero, J. B. M., & Villa, J. E. (2023). Epidemiology of fungal plant diseases in the Philippines. Mycology in the Tropics, 189-212., v) Gitaitis, R., & Walcott, R. (2007). The epidemiology and management of seedborne bacterial diseases. Annu. Rev. Phytopathol., 45(1), 371-397., vi) Llompart, M., Cifre, J., Olmo, D., Juan, A., Castellà, F., Jiménez, S., & Sabaté, J. (2025). Epidemiology of' Xylella fastidiosa'in Ibiza and Formentera: A Comprehensive Study of Insect Vectors and Transmission Dynamics. Agronomy, 2025, vol. 15, num. 2, 329.
In this context, we consider that the word epidemiology can be used.
Line 2: Since the manuscript does not show the involvement of insects, how appropriate is it to use "Insect" in the title of the manuscript? I find the title of the manuscript incorrect.
It is true that insect information was not generated in this paper. However, the 16S rRNA gene sequences obtained from infected trees, grass and weeds were compared with insect and plant sequences previously reported by us in GenBank. A paragraph was included in the text to clarify this point.
Taking into account that the involvement of the insects in the paper is small, the title was changed to “The role of grass in the epidemiology of a phytoplasma disease affecting trees and other plants of the Sabana de Bogotá, Colombia”
Lines 91-93: "40 trees were selected at random" - please clarify this phrase. Was the presence of signs of disease caused by phytoplasmas already taken into account during the selection?
Answer: In the text, we introduced a small explanation about the “random” selection of trees; for the symptom survey, the trees were selected regardless of the presence or absence of phytoplasmas. For PCR tests, we selected symptomatic trees. This was also clarified in the text.
Were the same number of trees selected in each location?
Answer: We explained in materials and methods that 10 trees were surveyed in each location.
What is the size of the populations of the studied plants in the selected locations?
Answer: The population of trees in each area is unknown. It is not clear who planted the trees analysed in this study, and as far as we know, there are no records of the number of existing trees, especially in the rural locations.
That is, how representative are the 40 trees for the studied populations?
Answer: Probably not. However, the intention of this project was to confirm the presence of phytoplasmas in the diseased trees, not to study the prevalence of the disease in the Sabana de Bogota. Any way, our data provide a “small” insight into the distribution of the pathogen in the area, which confirms our observations indicating that the disease is widely distributed in Sal. humboldtiana and Sa. nigra, as well as in many other tree species that have not been studied.
Line 182 and following: The authors use "S." to denote different genera of plants: Salix, Sambucus, Senecio, Sonchus. This is very inconvenient for perceiving the manuscript information. Such a remark for "L.": both Liquidambar and Lipidium; for "T.": both Trifolium and Taraxacum.
To help the readability of the text, the short versión of the following scientific names was changed to: Cenchrus clandestinus to Ce. Clandestinus: Salix humboldtiana to Sa. humboldtiana; Sambucus nigra to Sa. nigra; Amaranthus dubius to Am. dubius; Cymbalaria muralis to Cy. muralis; Fumaria capreolata to Fu. capreolata; Gnaphalium cheiranthifolium to Gn. cheiranthifolium; Gnaphalium spicatum to Gn. spicatum; Lepidium bipinnatifidum to Le. bipinnatifidum; Megathyrsus maximus to Me. maximus; Myosotis sylvatica to My. Sylvatica: Plantago major to Pl. major; Senecio vulgaris to Se. Vulgaris; Sonchus oleraceus to So. Oleraceus; Spergula arvensis to Sp. arvensis; Taraxacum officinale to Ta. Officinale.
Lines 185 and 379: "70%" - this value does not correspond to the data in Table 2.
The value in line 379 was corrected to 82,5%. During
And if only trees with symptoms of the disease were selected, why did not 100% of the trees have signs of the disease.
In materials and methods, we explained that for the symptom survey, we selected the trees of each species at random; that is, we did not sample only diseased trees. Therefore, we can determine the number of diseased trees from the total sample to produce a prevalence estimation. We did not test non-symptomatic trees, but there might be infected trees that do not show symptoms either because their phytoplasma titer might be low or they have some resistance to the pathogen.
Tables 1 and 2: Specify the number of trees from each location.
The number of samples was introduced in the table.
Lines 235-236: Forty trees of each species were analyzed, but 22 Salix humboldtiana and 16 Sambucus nigra samples were PCR detected for phytoplasmas. Why? How were samples selected for PCR analysis?
We initially conducted a symptom survey on 40 trees of each species, of these 22 Salix humboldtiana and 16 Sambus nigra had symptoms suggesting the presence of phytoplasmas. Samples of these 16 and 22 trees were collected and tested for phytoplasmas by nested PCR of the 16S rRNA gene. This information was corrected with a new sentence in the materials and methods section.
Lines 257-284 (discussion of Table 3): Table 3. What does "Especies" mean?
The mistake in table 3 was corrected.
For the prevention of diseases caused by phytoplasmas, it is interesting to discuss the plants in which phytoplasmas were not detected.
Few ideas about this subject were included in the discussion.
References
Christensen, N. M., Nicolaisen, M., Hansen, M., & Schulz, A. (2004). Distribution of phytoplasmas in infected plants as revealed by real-time PCR and bioimaging. Molecular Plant-Microbe Interactions, 17(11), 1175-1184.
Herath, P., Hoover, G. A., Angelini, E., & Moorman, G. W. (2010). Detection of elm yellows phytoplasma in elms and insects using real-time PCR. Plant Disease, 94(11), 1355-1360.
Osman, F., and Rowhani, A. 2006. Applicationof a spotting sample preparation technique forthe detection of pathogens in woody plants byRT-PCR and real-time PCR (TaqMan). J. Virol.Methods 133:130-136
Siddique, A. B. M., Guthrie, J. N., Walsh, K. B., White, D. T., & Scott, P. T. (1998). Histopathology and within-plant distribution of the phytoplasma associated with Australian papaya dieback. Plant disease, 82(10), 1112-1120.
Round 2
Reviewer 1 Report
Comments and Suggestions for Authors
The authors of the manuscript have addressed all of my comments. The revised version of the manuscript is significantly improved compared to the previous version. I recommend that the manuscript be accepted for publication. Congratulations to the authors for an excellent job.